# Exploratory Analysis of Outpatient Visits for US Adults Diagnosed with Lupus Erythematosus: Findings from the National Ambulatory Medical Care Survey 2006–2016

**DOI:** 10.3390/healthcare10091664

**Published:** 2022-08-31

**Authors:** Salena Marie Preciado, Khaled A. Elsaid, Souhiela Fawaz, Lawrence Brown, Enrique Seoane-Vazquez, Marc Fleming, Yun Wang

**Affiliations:** 1Department of Pharmaceutical Economics and Policy, School of Pharmacy, Chapman University, Irvine, CA 92618, USA; 2Department of Biomedical and Pharmaceutical Sciences, School of Pharmacy, Chapman University, Irvine, CA 92618, USA

**Keywords:** systemic lupus erythematosus, prescription patterns, medications, comorbidities, retrospective study, autoimmune diseases

## Abstract

The study aims to assess office-based visit trends for lupus patients and evaluate their medication burden, chronic conditions, and comorbidities. This cross-sectional study used data from the National Ambulatory Medical Care Survey (NAMCS), a survey sample weighted to represent national estimates of outpatient visits. Adult patients diagnosed with lupus were included. Medications and comorbidities that were frequently recorded were identified and categorized. Descriptive statistics and bivariate analyses were used to characterize visits by sex, age, race/ethnicity, insurance type, region, and reason for visit. Comorbidities were identified using diagnosis codes documented at each encounter. There were 27,029,228 visits for lupus patients from 2006 to 2016, and 87% them were on or were prescribed medications. Most visits were for female (88%), white (79%), non-Hispanic (88%) patients with private insurance (53%). The majority of patients were seen for a chronic routine problem (75%), and 29% had lupus as the primary diagnosis. Frequent medications prescribed were hydroxychloroquine (30%), prednisone (23%), multivitamins (14%), and furosemide (9%). Common comorbidities observed included arthritis (88%), hypertension (25%), and depression (13%). Prescription patterns are reflective of comorbidities associated with lupus. By assessing medications most frequently prescribed and comorbid conditions among lupus patients, we showcase the complexity of disease management and the need for strategies to improve care.

## 1. Introduction

Lupus erythematosus is a chronic multisystem autoimmune disease with variable clinical manifestations that include widespread inflammation in multiple organs. The United States (U.S.) has the highest recorded estimate of prevalence of lupus, which was discovered to be 241 per 100,000 persons [1,2]. Mortality rates of lupus have improved over the years [3]. However, reported rates of survival still remain half that of the aged-matched general population, ranging from 50% to 90%. There is a greater increase in risk among younger and ethnically diverse women [4,5], and there are significant geographic variations [6]. Lupus is typically managed with a combination of corticosteroids, anti-malarial agents, immunosuppressants, biologics, and non-steroidal anti-inflammatory drugs (NSAIDs). These medications help prevent disease flare-ups; however, successful lupus care is frequently associated with substantial drug-induced toxicity and treatment-related side effects, often leading to organ damage, infectious complications, and treatment-associated comorbidities [7,8].

With longer disease duration, treatment goals for lupus have changed. Attention is being paid to the reduction in therapy-related side effects and comorbidities [9,10]. The burden of comorbidities has been established in the literature [11,12]. Findings suggest that lupus patients have an increased risk of coexisting conditions, such as cardiovascular disease, osteoporosis, malignancies, and pulmonary disorders [13]. Clinical guidelines recommend monitoring these comorbid conditions proactively by starting preventative treatments when necessary [13], although the presence of these other diseases may complicate the disease’s course and treatment plan [14].

The European League Against Rheumatism (EULAR) and the American College of Rheumatology (ACR) have published recommendations for the management of lupus erythematosus over the years [15,16]. These recommendations provide guidance for pharmacological treatment, the management of manifestations, and adjunct therapy. Improvements in research and patient care have modified the current treatment landscape, and therapy typically involves a wide range of drugs, including immunosuppressants, corticosteroids, antimalarials, non-steroidal anti-inflammatory drugs (NSAIDs), and biological agents [10]. The long-term use of these therapies can cause substantial morbidities, including gastrointestinal conditions, infections, pulmonary complications, and cardiovascular disease. Patients are often on multiple therapeutics to manage lupus and its associated comorbidities; thus, polypharmacy is of concern among this population [17]. Additionally, patients’ multiple morbidities may impact optimal treatment selection for the management of lupus and adjunct therapy.

Clinical guidelines have evolved; however, general knowledge of the real-world management of lupus patients is lacking. Limited studies exist on medication utilization. Recent manuscripts have been published on the Asia Pacific Lupus Collaboration cohort [18], German patients, and a U.S. cohort of a specific population of lupus patients from a single center, which was not representative of the national population [19]. Another study from the U.S. from 1993 to 2010 demonstrated the diversity of medications prescribed by physicians to manage lupus but found that first-line treatment of lupus remained consistent over those years [20]. Novel therapeutics, such as biologics, have been approved for the treatment of lupus; thus, updated assessments of treatments prescribed and encountered comorbidities remain necessary. This study aimed to assess trends in visits for patients diagnosed with lupus and characterize the medication burden and the main comorbidities complicating the disease’s course.

## 2. Materials and Methods

Data were obtained from NAMCS from 2006 to 2016. The NAMCS by the Center for Disease Control and Prevention (CDC) and National Center for Health Statistics (NCHS) is nationally representative of medical office visits from non-federally employed office-based physicians in the U.S., including specialists and primary care providers [21]. Sample weights were included in the analysis to adjust for the multistage sampling design and survey nonresponse; detailed information is provided on the CDC website [22].

Visits of adult patients (≥18 years) with a diagnosis of lupus erythematosus and systemic lupus erythematosus (SLE) were captured using the International Classification of Diseases, Ninth and Tenth Revisions, clinical modification codes: ICD-9-CM (710.0, 695.4) and ICD-10-CM (M32, M32.0, M32.1, M32.8, M32.9, M32.10, L93, L93.0, L93.1, and L93.2) [23]. SLE is the most common type of lupus, but codes for lupus erythematosus (discoid and other) were also included for a more comprehensive sample. The study’s outcome variable was the medication prescribed or reported by the physician during the medical visit. Medications were identified using the CDC’s Ambulatory Care Drug Database system, which uses the Cerner Multum Lexicon Plus^®^ database (Cerner Multum Inc., Denver, CO, USA) for identifying medications based on NCHS 5-digit generic codes [24]. The drug categories were determined using Multum’s third level therapeutic categories (Appendix A). Information on up to 30 drugs and prescribing status was reported each visit. The dataset has up to five diagnosis codes listed per patient visit. Diagnosis codes were used to identify comorbidities using ICD-9-CM and ICD-10-CM codes and grouped by Multi-level Clinical Classification Software (CCS) categories provided by the Healthcare Cost and Utilization Project [23] (Appendix A). Patient demographics and clinical characteristics include age, sex, race, ethnicity, region, insurance type, reason for visit, chronic conditions, comorbidities, and medications prescribed.

National trends in outpatient visits for patients with lupus were assessed in the study period. Descriptive statistics for unweighted and weighted visits were calculated to examine characteristics of visits. Chi-square tests were performed to identify significant differences in visits between two consecutive time periods (2006–2010 and 2011–2016). Medications prescribed were compared to observe changes in therapy after the approval of belimumab, the first FDA approved biologic for treatment of lupus in 2011 [25]. Analyses were performed using R Programming, and significance was set to *p* < 0.05.

## 3. Results

During the study period, a total of 969 unweighted ambulatory visits, which corresponded to 27.18 million weighted visits, were identified for patients diagnosed with lupus. Visits varied by year, reaching a peak in 2011 (Appendix A)**.** More visits for women and visits in the South were observed when comparing 2006–2010 to 2011–2016, but this was not significant. Patient visits were greater among women (n = 23.79 million; 88%) (Table 1). Most patients were white (n = 21.31 million; 79%), non-Hispanic (n = 24.02 million; 88%), aged 45–59 (n = 10.62 million; 39%), had private insurance (n = 14.34 million; 53%), and were living in urban metropolitan statistical areas (n = 25.41 million; 93%) in the South (n = 10.36 million; 38%). Less than one-third of visits (n = 7.95 million; 29%) involved a patient with a recorded primary lupus-related diagnosis. Most patients were seen for a chronic problem (n= 20.46 million; 75%). Bivariate analysis showed significant differences in insurance type and the major reason for visit when comparing the two-year periods (*p* < 0.05). There was an average ± SD of 5 ± 1.3 medications reported per visit.

Arthritis was the most common chronic condition reported among lupus patients (n = 23.82 million, 88%). Other commonly reported chronic conditions included hypertension (n = 6.89 million; 25%), depression (n = 3.47 million; 13%), and hyperlipidemia (n = 2.87 million; 11%). Nearly half of the patients had one reported chronic condition (n = 13.04 million; 48%), and 21% had two (n = 5.78 million) (Table 2). Common comorbidities identified during lupus visits included diseases of the skin and subcutaneous tissue (n = 14.54 million; 40%), diseases of the musculoskeletal system and connective tissue (n = 5.41 million; 15%), and diseases of the circulatory system (n = 4.54 million; 12%) (Table 3). 

From 2006 to 2016, the most frequently prescribed medications by drug category were centrally nervous system (CNS) drugs (n = 13.33 million; 15%), hormones (n = 9.92 million; 11%), cardiovascular agents (n = 8.50 million; 10%), and drugs categorized as other (n = 9.64 million; 11%). Significant differences were found in the reported numbers of anti-infectives, nutritional products, and drugs classified as other over the years. There was an increase in those medications from 2006–2010 to 2011–2016 (Figure 1). Biologicals were prescribed minimally (n = 362,170; 0.04%). The top medications prescribed in visits for lupus patients from 2006 to 2010 were prednisone (n = 2.00 million; 19%), hydroxychloroquine (n = 1.91 million; 18%), multivitamins (n = 864,744; 8%), esomeprazole (n = 692,747; 7%), and methotrexate (n = 691,950; 7%) (Appendix A). From 2011 to 2016, the top medications prescribed were hydroxychloroquine (n = 6.22 million; 37%), prednisone (n = 4.15 million; 25%), multivitamins (n = 2.89 million; 17%), furosemide (n = 2.21 million; 13%), and folic acid (n = 1.47 million; 9%). Further, a significantly higher number of prescriptions for hydroxychloroquine than for other drugs was identified (*p* < 0.05).

## 4. Discussion

Lupus is predominantly seen among women, with a ratio of 9:1 compared to men. Our findings were consistent with this; however, the onset of lupus is commonly reported in women of childbearing age, whereas this analysis found that the majority of visits were among older patients aged 45–59 years. Although status of an initial diagnosis was not captured in this study, it is possible that it largely included women who were diagnosed at an earlier time, when the onset of the disease typically occurs. Another important finding was a significant increase in visits for women 45–59 years old in the study period. Although an older population likely with more comorbidities may be conducive to a greater number of medical visits, it only partially explains the increase observed in 2011–2016. Previous literature has reported late-onset lupus in women, which may be triggered by menopause or age-related changes to the immune system that impact cellular functions and may contribute to the development of lupus in older women [26,27].

The increase in visits seen after 2010 could also be related to other factors, such as the implementation of the Affordable Care Act (ACA) in 2010, which has had a substantial impact on access to healthcare [28]. The percentage of people without health insurance reached 16% in 2010 and declined to nearly 10% in 2016 [29]. The present study observed a much lower number of patients using self-pay as a form of payment in the second period: 5% in 2006–2010 decreased to 2% in 2011–2016. The introduction of ACA has made health care more accessible, particularly to low-income minorities, and may explain the significant association we observed between insurance type and time period [30].

Patients with lupus are three times more likely to suffer from multimorbidity [31], and our analysis showed that lupus patients experience a spectrum of comorbidities, including hypertension, arthritis, and diseases of the skin and subcutaneous tissue. A genetic link between arthritis and lupus has been found, and skin disorders such as systemic scleroderma are common [32]. These findings further confirm the increased disease burden among patients with lupus and the need for systematic screening for a range of comorbidities at diagnosis and throughout management of lupus. Significant differences were seen in the prescribing of medications during the study period, specifically among anti-infectives and drugs classified as other. Patients are often immunocompromised due to the treatment of their underlying disease, and this dysfunction of the immune system increases risk for infection in lupus patients, including bacterial, viral, and fungal infections [33]. Drugs classified as other largely consisted of antirheumatic drugs. This may be explained by the fact that the majority of patients had a chronic condition of arthritis, and there was an increase seen in 2011–2016 compared to 2006–2010.

The most widely prescribed medications for patients with lupus changed from prednisone (19%) in 2006–2010 to hydroxychloroquine (37%) in 2011–2016. This may be explained by the fact that hydroxychloroquine is now recommended in clinical guidelines for all lupus patients due to its ability to reduce flares [15] and because long-term corticosteroid use can result in progressive organ damage and increased disease burden [8]. Corticosteroids have been reported in earlier studies to be the most commonly used lupus-related medications, particularly in hospitals [34]. With the approval of biologics in 2011, it was expected to see a greater number of visits with biologics prescribed (0.04%). This study supports previous evidence of low biologics use since biologics were not among the top medications prescribed [34]. Generally, the uptake of newer medications takes some time in clinical practice, which may explain the limited biologics prescribing observed. Pharmacists can play a significant role in patient-care teams in chronic disease management, promoting the physician prescribing of appropriate medications if biologics are warranted. Additionally, medications associated with high costs (such as biologics) are often limited by tiered insurance coverage, thereby impacting patient access [35]. Due to the nature of survey data, it is less possible to link each patient’s insurance type to the coverage for each drug. We expect other national claims databases could help us better understand the biologics and insurance tier coverage. However, most claims databases only capture patients with commercial insurance or Medicare/Medicaid. Our study offers a concrete overview of all insurance types.

Medications taken by patients with lupus were also representative of their comorbid conditions. The most common drugs prescribed were hormones, CNS drugs, and cardiovascular agents. Drugs classified as hormones consist of steroids, specifically corticosteroids, which are one of the common classes of drugs used for lupus disease inflammation and immune response. While cardiovascular agents are relatively common due to the way lupus affects the cardiovascular system, by increasing the risk of hypertension, atherosclerosis, pericarditis, myocarditis, and endocarditis [36]. CNS drugs are frequently prescribed because of the neuropsychiatric manifestations that are secondary to the disease and the pain associated with disease flares [37]. A study among German patients reported the mean number of prescriptions increased to 9.5 in 2018, which is much higher than the number in this study [12]. However, it is important to recognize the vast differences between the two countries. The U.S. multi-payer healthcare system likely limits lupus patients’ access to treatment and certain therapies due to high costs. Patients with lupus often take several medications (an average of five reported in this study), increasing the risk of polypharmacy, which can lead to greater risks of serious adverse events and drug–drug interactions. Polypharmacy has also been found to be associated with treatment response in other rheumatic diseases [38]. Future research should evaluate the effects of polypharmacy on the risks of adverse events, hospitalization, and mortality in lupus. This may also be an opportunity for pharmacists to become integrated into lupus care by engaging in areas of drug monitoring, treatment-related adverse effects, and identifying appropriate medications.

There is currently no cure for lupus, although applying the correct treatment strategy can help. More drugs are being used off-label to manage the disease, and recent approvals have expanded the therapies available for lupus, which now include anifrolumab and voclosporin [39,40]. Future studies are necessary to determine how this may impact future medication utilization among lupus patients and their adherence to these medications whose route of administration is intravenous. While advances in treatment are being made, there continues to be demand for research to assess their impacts on patients’ lives and well-being.

The use of secondary data led to the typical inherent limitations of weighted, multilevel sampled national surveys. First, NAMCS is not a longitudinal study following individuals across each year of the survey. The unit of analysis for this study was patient visit associated with a lupus diagnosis. Patients who frequently visited their physicians were more likely to appear in the analysis. NAMCS did not provide disease severity and did not match each medication with a specific current diagnosis, which made it difficult to assert the reason for the use of each medication. Over-the-counter products purchased by the patient were not captured (i.e., NSAIDS), only medications ordered by the provider during the visit. Additionally, NAMCS does not provide detailed information on medication orders, such as dosage, quantity, or strength. Lastly, more recent data from NAMCS were not available to include for analysis due to delays brought on by the coronavirus pandemic. Despite these limitations, unbiased national estimates of office-based visits for patients with lupus were provided, which were not limited to commercially insured or Medicare/Medicaid patients.

## 5. Conclusions

In this nationally representative study, comorbidities, chronic conditions, and medications associated with lupus were identified. These findings helped us characterize the medication and disease burdens among lupus patients to better understand how to develop strategies to improve disease management. The data further highlighted the complex care needs of patients with lupus and the pressing need for the improved management of conditions secondary to lupus through preventative measures. Identifying trends in medication utilization and understanding the burden of co-morbid conditions on the outcomes and management of this disease can inform future recommendations for clinical practice and the treatment of lupus.

## Figures and Tables

**Figure 1 healthcare-10-01664-f001:**
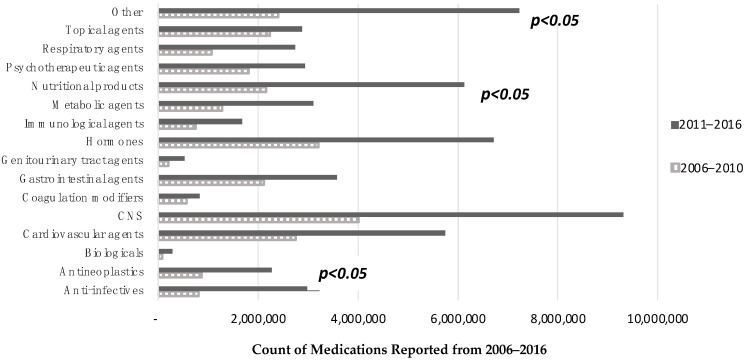
**Frequency of medications prescribed for patients diagnosed with lupus by drug category.** Drug categories were identified using the Cerner Multum Lexicon third level therapeutic category codes. Each p-value compares 2006–2010 and 2011–2016 and significance was set to <0.05. Abbreviations: CNS = central nervous system. Another drug category includes alternative medications, medical gases, miscellaneous agents, pharmaceutical aids, radiological agents, and plasma expanders.

**Table 1 healthcare-10-01664-t001:** Patient and visit characteristics of all patients diagnosed with lupus erythematosus.

	Total (n = 27,183,195)			2006–2010 (n = 10,378,266)		2011–2016 (n = 16,804,929)		
	Unweighted Visits	Weighted Visits (%)		Weighted Visits (%)		Weighted Visits (%)		*p*-Value ^e^
Sex								0.12
Male	140	3,389,897	(12%)	1,760,617	(17%)	1,629,280	(10%)	
Female	829	23,793,298	(88%)	8,617,649	(83%)	15,175,649	(90%)	
Age								0.11
18–29	77	2,182,239	(8%)	939,221	(9%)	1,243,018	(7%)	
30–44	213	5,458,507	(20%)	2,533,925	(24%)	2,924,582	(17%)	
45–59	333	10,619,135	(39%)	3,158,078	(30%)	7,461,057	(44%)	
60+	346	8,923,314	(33%)	3,747,042	3 (36%)	5,176,272	(31%)	
Race								0.10
White	770	21,381,927	(79%)	8,433,533	(81%)	12,948,394	(77%)	
Black	147	4,324,579	(16%)	1,134,157	(11%)	3,190,422	(19%)	
Other ^a^	52	1,476,689	(5%)	810,576	(8%)	666,113	(4%)	
Ethnicity								0.13
Hispanic	92	3,159,206	(12%)	822,696	(8%)	2,336,510	(14%)	
Non-Hispanic	877	24,023,989	(88%)	9,555,570	(92%)	14,468,419	(86%)	
Insurance								<0.05 *
Private	501	14,339,049	(53%)	5,222,761	(50%)	9,116,288	(54%)	
Medicare	272	6,991,175	(26%)	2,612,216	(25%)	4,378,959	(26%)	
Medicaid	84	2,352,709	(9%)	1,110,878	(11%)	1,241,831	(7%)	
Self-pay	31	732,993	(3%)	472,657	(5%)	260,336	(2%)	
N/A or blank ^c^	58	2,187,151	(8%)	484,884	(5%)	1,702,267	(10%)	
Other ^b^	30	580,118	(2%)	474,870	(5%)	105,248	(1%)	
Region								0.42
Northeast	148	5,013,587	(18%)	2,083,870	(20%)	2,929,717	(17%)	
Midwest	229	4,566,433	(17%)	2,314,255	(22%)	2,252,178	(13%)	
South	348	10,362,270	(38%)	3,320,615	(32%)	7,041,655	(42%)	
West	244	7,240,905	(27%)	2,659,526	(26%)	4,581,379	(27%)	
MSA Setting ^c^								0.26
Urban	886	25,405,771	(93%)	9,451,528	(91%)	15,954,243	(95%)	
Rural	83	1,777,424	(7%)	926,738	(9%)	850,686	(5%)	
Primary Diagnosis ^d^								0.57
Lupus related	248	7,945,471	(29%)	3,092,785	(30%)	4,852,686	(29%)	
Non-lupus related	448	19,237,724	(71%)	7,285,481	(70%)	11,952,243	(71%)	
Reported Any Medication								0.22
Yes	823	23,774,397	(87%)	8,795,251	(85%)	14,979,146	(89%)	
No	146	3,408,798	(13%)	1,583,015	(15%)	1,825,783	(11%)	
Major Reason for Visit								<0.05 *
Chronic problem	700	20,460,610	(75%)	7,787,237	(75%)	12,673,373	(75%)	
Acute or new problem	157	3,745,659	(16%)	1,477,435	(14%)	2,738,152	(16%)	
Preventative care	76	1,650,061	(6%)	686,467	(7%)	963,594	(6%)	
Other	36	856,937	(3%)	427,127	(4%)	429,810	(3%)	

^a^ Other race includes: Asian, Native Hawaiian or other Pacific Islander, American Indian or Alaskan Native, and more than one race reported. ^b^ Other insurance includes no charge/charity, and worker’s compensation; other major reasons for visit included pre- and post-surgery, and blank. ^c^ Abbreviations: N/A = not applicable; MSA setting = metropolitan statistical area. ^d^ Primary diagnosis refers to the provider’s primary diagnosis for this visit. ^e^ *p*-values obtained from Rao–Scott chi-square test of independence with second order adjustment; *p*-value compares 2006–2010 and 2011–2016. * Significance < 0.05.

**Table 2 healthcare-10-01664-t002:** Chronic conditions reported for lupus patient visits from 2006 to 2016.

	Total (n *=* 271,83,195)			2006–2010 (n = 10,378,266)		2011–2016 (n = 16,804,929)		
	Unweighted	Weighted (%) ^c^		Frequency (%)		Frequency (%)		*p*-Value ^d^
Chronic condition								
Arthritis	862	23,815,893	(88%)	8,971,767	(38%)	14,844,126	(62%)	0.68
Asthma	66	1,845,517	(7%)	642,601	(35%)	1,202,916	(65%)	0.77
Cancer	54	1,319,893	(5%)	238,165	(18%)	1,081,728	(82%)	<0.05 *
Cardiovascular Disease	35	1,038,823	(4%)	328,453	(32%)	710,370	(68%)	0.53
COPD ^a^	41	1,259,167	(5%)	650,490	(52%)	608,677	(48%)	0.27
Depression	137	3,473,664	(13%)	1,151,609	(33%)	2,322,055	(67%)	0.56
Diabetes	80	2,233,577	(8%)	975,686	(44%)	1,257,891	(56%)	0.53
Hyperlipidemia	104	2,871,639	(11%)	1,181,502	(41%)	1,690,137	(59%)	0.52
Hypertension	250	6,889,116	(25%)	2,270,024	(33%)	4,619,092	(67%)	0.47
Kidney disease	29	941,197	(3%)	477,509	(51%)	463,688	(49%)	0.34
Obesity	59	1,884,787	(7%)	490,665	(26%)	1,394,122	(74%)	0.27
Osteoporosis	62	2,183,236	(8%)	885,360	(41%)	1,297,876	(59%)	0.79
Other ^b^	30	594,703	(2%)	155,840	(26%)	438,863	(74%)	0.45
Total chronic conditions								0.39
0	54	1,474,049	(5%)	861,956	(58%)	612,093	(42%)	
1	423	13,043,442	(48%)	4,986,769	(38%)	8,056,673	(62%)	
2	242	5,784,626	(21%)	2,090,053	(36%)	3,694,573	(64%)	
3	140	3,516,536	(13%)	1,244,555	(35%)	2,271,981	(65%)	
4	74	2,050,399	(8%)	432,016	(21%)	1,618,383	(79%)	
5 or more	33	1,210,239	(4%)	664,996	(55%)	545,244	(45%)	

^a^ Abbreviations: COPD—chronic obstructive pulmonary disease. ^b^ Other chronic conditions include obstructive sleep apnea, HIV, cerebrovascular disease, and substance abuse. ^c^ Total chronic conditions may not add up to 100% due to N/As or missing values. ^d^ *p*-values obtained from Rao–Scott chi-square test of independence with second order adjustment; *p*-value compares 2006–2010 and 2011–2016. * Significance < 0.05.

**Table 3 healthcare-10-01664-t003:** Common comorbidities observed in lupus patient visits from 2006 to 2016 (identified by ICD-9 and ICD-10 codes).

	Total		2006–2010		2011–2016		
Comorbid Condition Categories	Unweighted	Weighted	Frequency (% ^b^)		Frequency (% ^b^)		*p*-Value ^c^
Infectious And Parasitic Diseases	77	2,431,997	2,062,607	(85%)	369,390	(15%)	<0.05 *
Neoplasms	34	949,063	329,488	(35%)	619,575	(65%)	0.88
Mental, Behavioral and Neurodevelopmental Disorders	75	1,603,369	881,809	(55%)	721,560	(45%)	0.18
Diseases of the Nervous System and Sense Organs	56	984,724	646,998	(66%)	337,726	(34%)	<0.05 *
Diseases Of the Circulatory System	139	4,539,925	1,718,098	(38%)	2,821,827	(62%)	0.87
Diseases Of the Respiratory System	51	1,876,549	489,284	(26%)	1,387,265	(74%)	0.34
Diseases of the Skin and Subcutaneous Tissue	496	14,538,032	5,462,559	(38%)	9,075,473	(62%)	0.78
Diseases Of the Musculoskeletal System and Connective Tissue	177	5,405,844	2,987,647	(55%)	2,418,197	(45%)	<0.05 *
Other Diseases ^a^	44	1,851,505	999,753	(54%)	851,752	(46%)	0.05
Symptoms, Signs, and Abnormal Clinical and Laboratory Findings, Not Elsewhere Classified	75	2,354,082	854,807	(36%)	1,499,275	(64%)	0.95

^a^ Other diseases include: endocrine, nutritional, metabolic diseases, immunity disorders, diseases of the blood and blood-forming organs, certain disorders involving the immune mechanisms, diseases of the digestive and genitourinary system, injury, poisoning, and certain other consequences of external causes. ^b^ Denominator for proportions is total visits for lupus patients for that category. ^c^ *p*-values obtained from Rao–Scott chi-square test of independence with second order adjustment; *p*-value compares 2006–2010 and 2011–2016. * Significance < 0.05.

## Data Availability

Data for national health surveys are publicly available on the CDC website and can be found at https://www.cdc.gov/nchs/ahcd/index.htm (accessed on 24 July 2022).

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
