# Peer review of "Exploratory Analysis of Outpatient Visits for US Adults Diagnosed with Lupus Erythematosus: Findings from the National Ambulatory Medical Care Survey 2006–2016"

_healthcare, 2022, doi:10.3390/healthcare10091664_

Round 1

Reviewer 1 Report

This study from Preciado and colleagues presents important research on the treatment patterns in Lupus patients. While prior studies have already reported the trends in SLE using NAMCS data, this study provides new updated information. I hope you find my comments useful. Some specific items for clarification are outlined:

Reviewer 2 Report

1.       Title should be revised as medication observed sounds something non-scientific.  

2.       Author’s affiliation must contain city and county name.

3.       Keywords can be modified as per title to improve search engine visibility upon publication.

4.       Why in Table 2 data is separated from 2006-2010 and 2011-2016? Was it only for the sake of comparison? If so what were the outcomes?

5.       Since the data was taken till 2016, some discussion on current situation on lupus patients medication consumption and adherence would be beneficial.
